

# The mediating role of attention deficit in relationship between insomnia and social cognition tasks among nurses in Saudi Arabia: A cross-sectional study

Md. Dilshad Manzar[1], Faizan Kashoo[2], Abdulrhman Albougami[1], Majed Alamri[3], Jazi Shaydied Alotaibi[1], Bader A. Alrasheadi[1], Ahmed Mansour Almansour[1], Mehrunnisha Ahmad[1], Mohamed Sherif Sirajudeen[2], Mohamed Yacin Sikkandar[4] and Mark D. Griffiths[5]

[1] Department of Nursing, College of Applied Medical Sciences, Majmaah Univerity, Al Majmaah, Riyadh, Saudi Arabia
[2] Department of Physical Therapy and Health Rehabilitation, College of Applied Medical Sciences, Majmaah University, Majmaah, Riyadh, Saudi Arabia
[3] Department of Nursing, College of Applied Medical Sciences, University of Hafr Al Batin, Hafr Al Batin, Saudi Arabia
[4] Department of Medical Equipment Technology, College of Applied Medical Sciences, Majmaah University, Majmaah, Riyadh, Saudi Arabia
[5] International Gaming Research Unit, Psychology Department, Nottingham Trent University, Nottingham, United Kingdom

Corresponding author
Md. Dilshad Manzar,
m.manzar@mu.edu.sa

## ABSTRACT

**Purpose:** Insomnia-related affective functional disorder may negatively affect social cognition such as empathy, altruism, and attitude toward providing care. No previous studies have ever investigated the mediating role of attention deficit in the relationship between insomnia and social cognition.

**Methods:** A cross-sectional survey was carried out among 664 nurses ($M_{age}$ = 33.03 years; SD ± 6.93 years) from December 2020 to September 2021. They completed the Scale of Attitude towards the Patient (SAtP), the Athens Insomnia Scale (AIS), a single-item numeric rating scale assessing the increasing severity of attention complaints, and questions relating to socio-demographic information. The analysis was carried out by examining the mediating role of attention deficit in the relationship between insomnia and social cognition.

**Results:** The prevalence of insomnia symptoms was high (52% insomnia using the AIS). Insomnia was significantly correlated with attention problems ($b$ = 0.18, standard error ($SE$) = 0.02, $p < 0.001$). Attention problems were significantly negatively correlated with nurses' attitudes towards patients (b = −0.56, SE = 0.08, $p < 0.001$), respect for autonomy (b = −0.18, SE = 0.03, $p < 0.001$), holism (b = −0.14, SE = 0.03, $p < 0.001$), empathy (b = −0.15, SE = 0.03, $p < 0.001$), and altruism (b = −0.10, SE = 0.02, $p < 0.001$). Attention problems indirectly mediated the effect of insomnia on attitudes toward patients (99% CI = −0.10 [−0.16 to −0.05]), respect for autonomy (99% CI = −0.03 [−0.05 to −0.02]), holism (99% CI = −0.02 [−0.04 to −0.01]) empathy (99% CI = −0.03 [−0.04 to −0.01]), and altruism (99% CI = −0.02 [−0.03 to −0.01]).

**Conclusion:** Nurses with insomnia-related attention problems are likely to have poor explicit social cognition such as attitude toward patients, altruism, empathy, respect for autonomy, and holism.

# INTRODUCTION

Insomnia is characterized by repeated difficulties with sleep onset, maintenance, consolidation and/or quality despite having enough time and opportunity to sleep (*Morin, 1993*). Working conditions, sleep, and mental health conditions are often interrelated among nurses. Currently, there is a serious shortage of healthcare professionals, including nurses, in 57 nations (*World Health Organization, 2016*). This is projected to cause an estimated global shortfall of 15 million healthcare professionals by 2030 (*World Health Organization, 2016*). Such shortages often lead to an increase in workload, as well as long and irregular shifts in duty schedules among nurses (*Eldevik et al., 2013*; *Jradi, Alanazi & Mohammad, 2020*; *Labrague et al., 2018*). Physical and mental stress among nurses is associated with an increase in workload brought on by a shortage of nurses (*Jradi, Alanazi & Mohammad, 2020*; *Labrague et al., 2018*).

Long and irregular shifts in duty schedules are often associated with insomnia, excessive daytime sleepiness, and psychosomatic problems (*Eldevik et al., 2013*). Insomnia and sleep loss are associated with attention problems (*Sving et al., 2012*), lower empathy (*Hobeika et al., 2020*), and decreased altruism (*Ben Simon et al., 2022*) among health professionals including nurses. However, previous studies have mostly tried to understand impact of sleep/sleep loss on attention, empathy, and altruism separately. Moreover, no studies have examined the inter-relationship between insomnia, attention deficit, and social cognitive aspects such as empathy and altruism in a single model.

## Healthcare delivery-related social cognitive tasks

Social cognition is the area of social psychology that explores social behavior, interactions, and activities (*Stangor, Jhangiani & Tarry, 2022*). *Pawlikowski, Sak & Marczewski (2012)* developed a tool to assess ethical attitudes toward patients among healthcare professionals. Attitudes refer to a set of emotions, beliefs, and behaviors toward a specific object, person, thing, or event (*Hobeika et al., 2020*). *Pawlikowski, Sak & Marczewski (2012)* tool has four factors comprising empathy, altruism, respect for autonomy, and holism. In relation to nursing and healthcare, (i) respect for autonomy implies that patients can make independent decisions; (ii) holism implies that nurses should consider a patient's body, mind, spirit, culture, socioeconomic background, and environment when delivering care; (iii) empathy implies understanding patients' emotions, feelings, and thoughts; and (iv) altruism implies selflessness while delivering patient care (*Pawlikowski, Sak & Marczewski, 2012*).

Empathy is an important aspect of social cognition (*Bosco et al., 2015*). In social psychology, altruism is thought to arise from a multifaceted foundational root, one of which is social cognition (*Mattis et al., 2009*). Since, social cognitive approaches help in the understanding of attitude formation and behavior patterns (*Stangor, Jhangiani & Tarry, 2022*), the two remaining factors of *Pawlikowski, Sak & Marczewski (2012)* ethical attitude tool (*i.e.*, respect for patients' autonomy and holistic attitude) are most plausibly explained by social cognitive perspective (*Stangor, Jhangiani & Tarry, 2022*). Therefore, in the present study, healthcare delivery-related social cognitive tasks were defined to comprise empathy, altruism, respect for autonomy, and holism (*Pawlikowski, Sak & Marczewski, 2012*; *Stangor, Jhangiani & Tarry, 2022*).

## Insomnia, attention, and poor social cognition

Decreased vigilant attention associated with insomnia symptoms most plausibly explains the insomnia-cognition relationship (*Dorrian, Rogers & Dinges, 2004*; *Lim & Dinges, 2008*). Sleep deprivation-related decrease in vigilant attention is expressed as delayed response, mistakes in commission, and expression of interacting circadian and homeostatic sleep drive on prolonged wakefulness (*Lim & Dinges, 2008*). Moreover, a recent meta-analysis found that sustained attention has a strong association with cognitive empathy among healthy people (*Yan et al., 2020*). Sleep deficiency and poor sleep quality (insomnia caused by sleep loss, sleep deprivation, sleep restriction, and sleep fragmentation) can lead to an affective dysfunction (*Dorrian, Rogers & Dinges, 2004*; *Lim & Dinges, 2008*; *Ben Simon et al., 2020*). This can result in problems of emotional processing (reactivity, evaluation/expression), as well as in problems in social and workplace discourses (*Ben Simon et al., 2020*).

Problems in emotion processing are evident in emotional unpredictability, irritability, and decreased positive mood (*Ben Simon et al., 2020*). Sleep loss leads to erroneous emotion differentiation (*Ben Simon et al., 2020*). Individual-level emotional processing coupled with attentional bias and/or decreased vigilant attention may express in problems of social and workplace discourse such as reduced prosocial behavior, and increased abusive behavior (*Dorrian, Rogers & Dinges, 2004*; *Lim & Dinges, 2008*; *Ben Simon et al., 2020*). This may explain conduct problems, reduced altruistic behavior, less respect for other's opinion/autonomy, holism, trust, and trustworthiness (*Dickinson & McElroy, 2017*; *Ben Simon et al., 2020*). The capacity of nurses to empathize and offer compassionate care, both of which are vital to their profession, may be influenced by their sleep deficiency and poor quality.

In summary, it appears that sleep deficiency, daytime sleepiness, and poor sleep quality (*i.e.*, insomnia) can lead to decreased vigilant attention and attentional bias, that may express itself as problems in attitude and social cognition. However, no previous study has ever investigated all three variables simultaneously in a single model (*i.e.*, insomnia, attention deficit, and social cognitive tasks). To address this knowledge gap, the present study investigated the interplay between these three variables. It was hypothesized that among nurses, attention deficit would mediate the relationship between insomnia and

attitude towards patients (and its factors such as empathy, altruism, respect for patient's autonomy, and holistic attitude).

## METHODS

### Participants and procedure

Nurses working in healthcare facilities in Saudi Arabia ($n = 644$) participated in a cross-sectional survey study from December 2020 to September 2021. Convenience sampling was used to collect the data. The target population was nurses enrolled in the master's program at the university, and nurses working with them at their healthcare facilities. The only inclusion criterion was the possession of a valid Saudi Commission of Health Specialties license at the time of the study. There were no exclusion criteria for those that met the inclusion criterion. Saudi and expatriate nurses of various nationalities were invited to participate in the present study using two different forms of data collection (online and offline).

In the offline data collection, nurses were given a hard copy of the survey to complete by trained data collectors. In the online data collection, a survey link was e-mailed and shared through social networking sites with participants. In the offline method, a simple explanation regarding the research purpose, procedure, and intended objectives was given to nurses. After this, the participant provided informed written consent of participation. In the online version, a written introductory note provided information regarding the research purpose, procedure, and intended objectives in simple language. In the online mode, nurses provided their consent by selecting 'yes' for an item seeking their consent to participate in the study. In both online and offline modes, nurses were informed about their right to withdraw, the voluntary nature of participation, non-involvement of risks, and policy and procedure to maintain strict personal data confidentiality. Contact information was shared with the participants, for communication with the research team for their prospective queries and suggestions. A total of 471 nurses responded online, and 193 nurses responded offline.

In the present study, all procedures followed the Helsinki Declaration, 2002, and ethical standards and norms for research with human participants. The research proposal was approved by the Human Ethics Committee, Ministry of Health, Saudi Arabia (Approval no. 9; H-05-FT-083). Participants were required to complete the (i) Scale of Attitude towards the Patient (SAtP; *Alamri, 2020*; *Pawlikowski, Sak & Marczewski, 2012*) (ii) a single-item visual analog scale assissing increasing severity of attention complaints, and (iii) Athens Insomnia Scale (AIS) (*Manzar et al., 2022a*; *Soldatos, Dikeos & Paparrigopoulos, 2000*). The authors had permission to use the AIS and SAtP instruments from the copyright holders. All the scales were in English because all participants were professional nurses with an adequate level of proficiency in the language.

### Measures

#### *Athens Insomnia Scale*

The AIS was used to assess sleep difficulties associated with insomnia symptoms as determined by the International Classification of Diseases-10th Revision (ICD-10)

(*Soldatos, Dikeos & Paparrigopoulos, 2000*). The AIS is widely used in research and clinical settings for assessing insomnia (*Manzar et al., 2022a*; *Sirajudeen et al., 2020*). The AIS has eight items that appraise the subjective account of two groups of symptoms: sleep-time-related complaints, and daytime complaints (*Soldatos, Dikeos & Paparrigopoulos, 2000*, *2003*). Each item of the AIS (*e.g.*, *"Sleepiness during the day"*) is scored on a scale of 0 (*none*) to 3 (*intense*). A higher score indicates greater severity of insomnia symptoms. The AIS has been validated in various populations including Asian demographics such as nurses in Saudi Arabia, and Indian occupational computer users (*Manzar et al., 2022a*; *Sirajudeen et al., 2020*). In the present study, a McDonald's omega of 0.85 (95% CI [0.82–0.87]) indicated very good internal consistency.

### The Scale of Attitude towards the Patient

The SAtP was used to assess health professionals' attitudes towards patients (*Pawlikowski, Sak & Marczewski, 2012*). The SAtP comprises four domains/sub-scales: respect for autonomy, holism, empathy, and altruism. The SAtP has seven items (*e.g.*, *"In my opinion, a patient is above all a suffering human being in need of help"*), all scored on a seven-point scale from 1 (*strongly disagree*) to 7 (*strongly agree*). Higher scores on the SAtP and its sub-scales indicate better attitude among health professionals (nurses in the present study) for patients. Altruism has one item whereas the other three dimensions have two items each. SAtP has adequate reliability, and factorial validity among Polish doctors (*Pawlikowski, Sak & Marczewski, 2012*). A recent study in Saudi Arabia reported adequate reliability for SAtP among nurses in Saudi Arabia (*Alamri, 2020*). In the present study, a McDonald's omega of 0.86 (95% CI [0.83–0.88]) indicated very good internal consistency.

### Attention deficit rating item

A 0–10 numeric rating item was used to assess the severity of attention deficit-related complaints using the question *"Do you have difficulties in paying attention?"*. Higher scores indicate greater severity of attention complaints. A similar single-item visual analog scale has been used to assess attention deficit among university students (*Manzar et al., 2020*). In that previous study, a cut-off score of five and above was used to screen university students with attention problems. Two groups based on this cut-off score of the single-item attention tool differed significantly across a scale to assess migraine severity. This established the known-group validity of the migraine screening tool (*Manzar et al., 2020*). Moreover, the use of single-item tools is often employed in clinical and research settings (*Appukuttan et al., 2015*; *Kido-Nakahara et al., 2015*; *Manzar et al., 2020*; *Reich et al., 2012*). Similar single-item tools have been employed to evaluate a variety of clinical disorders, including pruritus, attentional issues, and dental anxiety (*Appukuttan et al., 2015*; *Kido-Nakahara et al., 2015*; *Manzar et al., 2020*; *Reich et al., 2012*).

### Socio-demographic information

The survey also included a number of questions regarding socio-demographic information. Participants were asked about their gender (male, female, and prefer not to say), and open-ended questions asked for their age, weight, height, experience as nurse (in years), and daily work schedule (in hours).

## Statistical analysis

SPSS version 23.0 along with the PROCESS macros version 4.0 were used to analyze the data (*Hayes, 2013*; *Hayes & Coutts, 2020*). Participants' characteristics are presented as means, standard deviations, and percentages. A total of 664 nurses participated in the study. If there were person-level missing values, these were deleted. In the final sample ($n = 664$ minus $8 = 656$). A total of 0.41% of missing values were found, and these missing values were spread across 3.81% of participants for variables in the mediation model. There was no pattern in the missing values. Multiple imputations for five iterations were used to manage the missing values. PROCESS macros cannot handle the analysis for pooled imputed data. Therefore, the imputed dataset (fifth iteration) was used in the mediation analysis. Multiple imputations were performed using the default method in the SPSS 23.0. This option automatically scans the data and accordingly applies one of the two, monotone or fully conditional specification (FCS) approaches.

Mersenne twister was the random number generator method with a fixed seeding of 200,000. In this default method, logistic regression (categorical variables), and linear regression (scale variables) are employed. Mediation analysis was performed employing a modified form of Baron and Kenny's method (*Baron & Kenny, 1986*; *Zhao, Lynch & Chen, 2010*). The mediating effect of attention deficit on the relationship between insomnia and nurses' attitudes toward patients (and its four dimensions, *i.e.*, respect for autonomy, holism, empathy, and altruism) was evaluated by analyzing five separate models.

Total score on the AIS was the predictor variable, while the numeric-rating scale of attention problems was the mediating variable. The outcome variables were the SAtP score (overall attitude) and its four sub-scale scores (respect for autonomy, holism, empathy, and altruism). Preacher and Hayes' criterion was used to determine the indirect effect. An absence of zero from the confidence interval of the bootstrapped coefficients implies that the indirect relationship is significant (*Preacher & Hayes, 2004*). As five mediation models were implemented on the dataset, a case of multiple testing, therefore, there was a $p$-value adjustment (*i.e.*, adjusted $p$-value = 0.05/5 (number of comparisons, in this case, mediation models), and confidence interval (CI) adjustment (*i.e.*, changed $p$-value from 95% to 99% for bootstrap confidence intervals) were used to establish significance level. Furthermore, to ensure the replicability of coefficients of the mediation models, seeding was fixed at 5,234 bootstraps by editing syntax of the PROCESS 4.0. Sleep deprivation-related attention process changes are age-dependent (*Tavakoli et al., 2023*). Obesity is a factor in the decrease in attention in sleep-deprived people (*Lucassen et al., 2014*). Previous research considered experience in years and working hours as covariates in determining the effect of sleep deprivation on psychomotor functions in nurses (*Johnson, Brown & Weaver, 2010*). Therefore, in the present study, age, BMI, nurses' experience in years, and duty hours every day were used as covariates.

**Table 1 Participants' characteristics of nurses.**

| Characteristics | Mean ± SD/Frequency (Percentage) |
| --- | --- |
| Age | 33.03 ± 6.93 |
| Gender | |
| Female | 485 (73.9) |
| Male | 164 (25.0) |
| Prefer not to say | 7 (1.1) |
| Body mass index (kg/m$^2$) | 25.44 ± 5.40 |
| Experience as nurse (in years) | 9.55 ± 6.46 |
| Daily duty hours | 8.48 ± 1.07 |
| Attention problem (on a scale of 1 to 10) | 3.26 ± 2.48 |
| Insomnia | |
| AIS score | 6.41 ± 4.75 |
| Insomnia symptoms | |
| Yes | 341 (52.0) |
| No | 311 (47.4) |
| Did not report | 4 (0.6) |
| SAtP score | |
| Total score (overall attitude towards patients) | 26.53 ± 4.76 |
| Respect for autonomy | 7.70 ± 1.52 |
| Holism | 7.51 ± 1.67 |
| Empathy | 7.52 ± 1.56 |
| Altruism | 3.80 ± 0.94 |

Note:
SD, standard deviation; AIS, Athens Insomnia Scale; SAtP, Scale of Attitude towards the Patient. Respect for autonomy. Holism, Empathy, and Altruism was assessed by factors of the SAtP scale. Presence/absence of insomnia symptoms was screened based on the AIS total score.

## RESULTS

### Participants' characteristics

The mean values for the participant characteristics were: age (33.03 years; SD ± 6.93), body mass index (25.44 kg/m$^2$; SD ± 5.40), experience as a nurse (9.55 years; SD ± 6.46), and daily work hours (8.48 h; SD ± 1.07) (Table 1). Prevalence of insomnia symptoms was high with 52% of participating nurses scoring 6 or higher on the AIS. The majority of the participants were females (73.9%). The mean and standard deviation of study variables were: attention-related problems (3.26 ± 2.48), SAtP total score (26.53 ± 4.76), respect for autonomy (7.70 ± 1.52), holism (7.51 ± 1.67), empathy (7.52 ± 1.56), and altruism (3.80 ± 0.94) (Table 1).

### Attention problems as a mediator in the effect of insomnia on nurses' attitudes towards patients: indirect-only mediation

Insomnia was significantly correlated with attention problems (b = 0.18, standard error (SE) = 0.02, $p < 0.001$), indicating those with severe insomnia complaints were likely to have more attention problems (Table 2). Attention problems were correlated with nurses'

**Table 2 Mediating role of attention problem through insomnia on attitude towards patients in nurses.**

| Independent variable | Outcome variable | β | b | SE | 99% bootstrapping CI | | p-value |
|---|---|---|---|---|---|---|---|
| | | | | | LL | UL | |
| Insomnia | Attention problem | 0.34 | 0.18 | 0.02 | 0.13 | 0.23 | <0.001 |
| Attention problem | Nurses' attitude towards patients | −0.29 | −0.56 | 0.08 | −0.76 | −0.36 | <0.001 |
| Insomnia (direct effect) | Nurses' attitude towards patients | 0.10 | 0.10 | 0.04 | −0.01 | 0.20 | 0.02 |
| **Types of effect** | **b** | | **SE** | | **99% bootstrapping CI** | | **p-value** |
| | | | | | LL | UL | |
| Total effect | −0.004 | | 0.04 | | −0.11 | 0.10 | 0.914 |
| Indirect effect | −0.10 | | 0.02 | | −0.16 | −0.05 | – |

Note:
Age (in years), body mass index (kg/m$^2$), experience as nurses (in years), and duty hours/day were used as covariates but none had any significant association (at adjusted $p < 0.01$). Insomnia was assessed by AIS score, and Nurses' attitudes towards patients was assessed by SAtP score. AIS, Athens Insomnia Scale; SAtP, Scale of Attitude towards the Patient; LL, lower limit; UL, Upper limit; SE, standard error; β, standardized coefficients; b, unstandardized coefficients; CI, confidence interval.

**Table 3 Mediating role of attention problem through insomnia on respect for patient's autonomy in nurses.**

| Independent variable | Outcome variable | β | b | SE | 99% bootstrapping CI | | p-value |
|---|---|---|---|---|---|---|---|
| | | | | | LL | UL | |
| Insomnia | Attention problem | 0.34 | 0.18 | 0.02 | 0.13 | 0.23 | <0.001 |
| Attention problem | Respect for patient's autonomy | −0.29 | −0.18 | 0.03 | −0.24 | −0.12 | <0.001 |
| Insomnia (direct effect) | Respect for patient's autonomy | 0.07 | 0.02 | 0.01 | −0.01 | 0.06 | 0.09 |
| **Types of effect** | **b** | | **SE** | | **99% bootstrapping CI** | | **p-value** |
| | | | | | LL | UL | |
| Total effect | −0.01 | | 0.01 | | −0.04 | 0.02 | 0.414 |
| Indirect effect | −0.03 | | 0.01 | | −0.05 | −0.02 | – |

Note:
Age (in years), body mass index (kg/m$^2$), experience as nurses (in years), and duty hours/day were used as covariates but none had any significant association (adjusted $p$-value $< 0.01$). Insomnia was assessed by AIS score, and Nurses' respect for patient's autonomy was assessed by SAtP factor score. AIS, Athens Insomnia Scale; SAtP, Scale of Attitude towards the Patient; LL, lower limit; UL, Upper limit; SE, standard error; β, standardized coefficients; b, unstandardized coefficients; CI, confidence interval.

attitudes towards patients (b = −0.56, SE = 0.08, $p < 0.001$), indicating those with attention problems were likely to have a poor attitude towards patients. The direct effect of insomnia on nurses' attitudes toward patients was not significant (considering an adjusted $p$-value of 0.01) after controlling for the effect of attention problems. However, the indirect effect was significant: 99% CI = −0.10 [−0.16 to −0.05], indicating that attention problems were a significant mediator in the relationship between insomnia and nurses' attitudes towards patients (Table 2). In summary, nurses with insomnia-related attention problems had poor attitudes toward patients.

### Attention problems as a mediator in the effect of insomnia on nurses' respect for patient's autonomy: indirect-only mediation

Attention problems were correlated with nurses' respect for patients' autonomy (b = −0.18, SE = 0.03, $p < 0.001$), indicating nurses with attention problems were likely to have poor respect for patients' autonomy (Table 3). The indirect effect was significant: 99%

**Table 4 Mediating role of attention problem through insomnia on nurses' holistic attitude towards patients.**

| Independent variable | Outcome variable | β | b | SE | 99% bootstrapping CI | | p-value |
|---|---|---|---|---|---|---|---|
| | | | | | LL | UL | |
| Insomnia | Attention problem | 0.34 | 0.18 | 0.02 | 0.13 | 0.23 | <0.001 |
| Attention problem | holistic attitude to patients | −0.20 | −0.14 | 0.03 | −0.21 | −0.06 | <0.001 |
| Insomnia (direct effect) | holistic attitude to patients | 0.09 | 0.03 | 0.02 | −0.005 | 0.07 | 0.028 |

| Types of effect | b | | SE | 99% bootstrapping CI | | p-value |
|---|---|---|---|---|---|---|
| | | | | LL | UL | |
| Total effect | 0.01 | | 0.01 | −0.03 | 0.04 | 0.585 |
| Indirect effect | −0.02 | | 0.01 | −0.04 | −0.01 | – |

Note:
Age (in years), body mass index (kg/m$^2$), experience as nurses (in years), and duty hours/day were used as covariates but none had any significant association (at adjusted $p < 0.01$). Insomnia was assessed by AIS score, and Nurses' holistic attitude towards patients was assessed by SAtP factor score. AIS, Athens Insomnia Scale; SAtP, Scale of Attitude towards the Patient; LL, lower limit; UL, Upper limit; SE, standard error; β, standardized coefficients; b, unstandardized coefficients; CI, confidence interval.

CI = −0.03 [−0.05 to −0.02], indicating that the attention problems mediated the relationship between insomnia and nurses' respect for patients' autonomy. In summary, nurses with insomnia-related attention problems had less respect for patients' autonomy.

### Attention problems as a mediator in the effect of insomnia on nurses' holistic attitude to patients: indirect-only mediation

Attention problems were correlated with nurses' holistic attitude towards patients (b = −0.14, SE = 0.03, $p < 0.001$), indicating those with attention problems were likely to have poor holistic attitude (Table 4). The direct effect of insomnia on nurses' holistic attitude to patients was not significant (considering an adjusted $p$-value of 0.01). The indirect effect was significant: 99% CI = −0.02 [−0.04 to −0.01], indicating that attention problems mediated the relationship between insomnia and nurses' holistic attitude to patients (Table 4). In summary, nurses with insomnia-related attention problems had a lower holistic attitude toward patients.

### Attention problems as a mediator in the effect of insomnia on nurses' empathy with patients: indirect-only mediation

Attention problems were correlated with nurses' empathy with patients (b = −0.15, SE = 0.03, $p < 0.001$), indicating those with attention problems were likely to have poor empathy (Table 5). The indirect effect was significant: 99% CI = −0.03 [−0.04 to −0.01], indicating that attention problems mediated the relationship between insomnia and nurses' empathy with patients. In summary, nurses with insomnia-related attention problems had lower empathy with patients.

### Attention problems as a mediator in the effect of insomnia on nurses' altruistic behavior towards patients: indirect-only mediation

Attention problems were correlated with nurses' altruistic behavior towards patients (b = −0.10, SE = 0.02, $p < 0.001$), indicating those with attention problem were likely to have poor altruism (Table 6). The direct effect of insomnia on nurses' altruistic behavior

**Table 5 Mediating role of attention problem through insomnia on nurses' empathy towards patients.**

| Independent variable | Outcome variable | β | b | SE | 99% bootstrapping CI | | p-value |
|---|---|---|---|---|---|---|---|
| | | | | | LL | UL | |
| Insomnia | Attention problem | 0.34 | 0.18 | 0.02 | 0.13 | 0.23 | <0.001 |
| Attention problem | Empathy towards patients | −0.24 | −0.15 | 0.03 | −0.21 | −0.08 | <0.001 |
| Insomnia (direct effect) | Empathy towards patients | 0.07 | 0.02 | 0.01 | −0.01 | 0.06 | 0.10 |
| **Types of effect** | **b** | | **SE** | | **99% bootstrapping CI** | | **p-value** |
| | | | | | LL | UL | |
| Total effect | −0.005 | | 0.01 | | −0.04 | 0.03 | 0.728 |
| Indirect effect | −0.03 | | 0.01 | | −0.04 | −0.01 | – |

Note:
AIS, Athens Insomnia Scale; Empathy towards patients is a factor of the SAtP scale; SAtP, Scale of Attitude towards the Patient. Age (in years), body mass index (kg/m$^2$), and duty hours/day were used as covariates but none had any significant association (at adjusted $p < 0.01$). Insomnia was assessed by AIS score, and Nurses' Empathy towards patients was assessed by SAtP factor score. AIS, Athens Insomnia Scale; SAtP, Scale of Attitude towards the Patient; LL, lower limit; UL, Upper limit; SE, standard error; β, standardized coefficients; b, unstandardized coefficients; CI, confidence interval.

**Table 6 Mediating role of attention problem through insomnia on altruistic behavior towards patients.**

| Independent variable | Outcome variable | β | b | SE | 99% bootstrapping CI | | p-value |
|---|---|---|---|---|---|---|---|
| | | | | | LL | UL | |
| Insomnia | Attention problem | 0.34 | 0.18 | 0.02 | 0.13 | 0.23 | <0.001 |
| Attention problem | Empathy towards patients | −0.26 | −0.10 | 0.02 | −0.14 | −0.06 | <0.001 |
| Insomnia (direct effect) | Empathy towards patients | 0.10 | 0.02 | 0.01 | 0.00 | 0.04 | 0.012 |
| **Types of effect** | **b** | | **SE** | | **99% bootstrapping CI** | | **p-value** |
| | | | | | LL | UL | |
| Total effect | 0.003 | | 0.01 | | −0.02 | 0.02 | 0.723 |
| Indirect effect | −0.02 | | 0.004 | | −0.03 | −0.01 | – |

Note:
Age (in years), body mass index (kg/m$^2$), experience as nurses (in years), and duty hours/day were used as covariates but none had any significant association (at adjusted $p < 0.01$). Insomnia was assessed by AIS score, and altruistic behavior towards patients was assessed by SAtP factor score. AIS, Athens Insomnia Scale; SAtP, Scale of Attitude towards the Patient; LL, lower limit; UL, Upper limit; SE, standard error; β, standardized coefficients; b, unstandardized coefficients; CI, confidence interval.

towards patients was not significant (b = 0.02, SE = 0.01, $p > 0.01$, *i.e.*, adjusted $p$-value of 0.01). Furthermore, 99% CI of the coefficients also established that there was no direct relationship (*Du Prel et al., 2009*). The indirect effect was significant: 99% CI = −0.02 [−0.03 to −0.01], indicating that attention problems mediated the relationship between insomnia and altruism (Table 6). In summary, nurses with insomnia-related attention problems had lower altruistic behavior toward patients.

# DISCUSSION

To the best of the authors' knowledge, the present study is the first to identify an indirect mediating effect of attention problems in the relationship between insomnia and nurses' overall attitude towards patients (as well as dimensions of social cognition such as empathy, holism, altruistic behavior towards patients, and respect for patient's autonomy). One of the novel findings in the present study was that there was no significant direct relationship between insomnia and attitude towards patient care (and dimensions), but the

relationship was indirectly mediated by attention problems. Moreover, there were no effects of covariates (age, BMI, duty hours/day, experience in years) in the mediation models.

The functional neuroanatomy of attention (of self and the individuals around them), and social cognition share common links. Moreover, the development of attention processing precedes social cognitive ability during the developmental stages (*Mundy & Newell, 2007*). The findings suggest that insomnia seems to disturb this functional balance between attention-social cognition dialogue. As outlined earlier in the paper, attitude, empathy, altruism, autonomy, and holism are all aspects of social cognition. Attitude has implicit and explicit features, with attention necessary for the expression of explicit features of attitude (*Greenwald & Banaji, 1995*). Moreover, functional neuroanatomy of attention and explicit features of social cognition seem to overlap in the anterior part of the medial prefrontal cortex, and in the lateral prefrontal cortex (*Van Overwalle, 2009*). Therefore, insomnia-related attention problems may be associated with the expression of explicit features of attitude and/or other social cognition through modulation of activity in the medial prefrontal cortex, and lateral prefrontal cortex. Additional studies are needed to further explore and establish the neuroanatomy of this relationship. Longitudinal studies with functional magnetic resonance imaging with discreet and non-overlapping tasks of explicit social cognition, and attention in individuals with/without insomnia and/or sleep deprivation models may help in better understanding the relationship dynamics of explicit social cognition, executive functions (*e.g.*, attention), and sleep.

Findings of the present study showed that attention problems mediated the relationship between insomnia and the nurse's attitudes toward patients. This implies that nurses with insomnia were more likely to suffer from attention problems. After controlling for the effect of attention problems, the direct relationship between insomnia and a nurse's attitudes toward patients was not significant. This implies that an intervention directed at improving attention problems among nurses is likely to improve nurses' attitudes toward patients (*Luo et al., 2021*).

The present study also found that attention problems mediated the relationship between insomnia and the nurse's respect for the patient's autonomy. This implies that a nurse with insomnia symptoms concurrently with attention problems had less respect for the patient's autonomy. A small study conducted among 79 Iranian nurses reported the moderate autonomy of patients during nursing care, whereas patients believed that their autonomy was not respected (*Rahmani, Ghahramanian & Alahbakhshian, 2010*). If a similar mediation effect is also established in longitudinal studies among healthcare professionals, then the management of insomnia and associated attention problems may be an important target modifiable factor to help improve patient outcome.

In the present study, attention problems mediated the relationship between insomnia and the nurse's holistic attitude to patients, empathy, and altruistic behavior. This implies that nurses with insomnia-related attention problems had a lower holistic attitude toward patients and poor empathy. Nurses with insomnia symptoms concurrently with attention problems had lower altruistic behavior towards patients. In this context, it is encouraging to note that, healthy sleep habits are one of the most commonly used coping strategies by

nurses as means of the health promotion model. This model is one of the most commonly used to implement methods and strategies of health promotion (*Lubinska-Welch et al., 2016*). Therefore, if these healthy sleep habits are implemented by nurses effectively, then these might help in managing the insomnia-related attention problems that affect explicit features of social cognition such as holistic attitude towards patients, empathy, and altruistic behavior (*Greenwald & Banaji, 1995*).

### Limitations

The present study had a number of limitations. A cross-sectional study cannot determine causal relationships. Therefore, a large-scale longitudinal study is required to determine causal relationships between the variables examined in the present study. However, as many participants have insomnia symptom-related attention problems, and/or social cognition deficits, involving them in longitudinal studies may limit their access to care. Such a situation raises uncomfortable ethical dilemmas (*Burnard et al., 2008*). In such circumstances, cross-sectional mediation analysis offers a more morally conscientious method of acquiring empirical data on mediating variables without preventing individuals from potentially receiving therapy for their attention issues associated with insomnia and/or social cognition deficiencies (*Manzar et al., 2021*; *Manzar et al., 2022b*; *Zhao, Lynch & Chen, 2010*). Functional magnetic resonance imaging studies with discreet and non-overlapping tasks of explicit social cognitions, attention among individuals with/without insomnia may also help in better understanding the functional neuroanatomy of both processes.

Moreover, data collected in the present study were self-reported by the participants, which may have influenced the results due to recall and social desirability bias. All the participants were from KSA, so the findings may not necessarily be generalizable to nurses from other countries. Therefore, the present study should be replicated in other countries and cultures. Finally, an important limitation is that the attention deficit was assessed using a single-item tool. Previous studies have shown the applicability, validity, and advantage of single items tools. Nevertheless, future studies with psychometrically robust measures may help to better understand the relationship between insomnia, attention deficit, and social cognition.

## CONCLUSION

Nurses with insomnia-related attention problems are likely to have poor explicit social cognition such as attitude toward patients, altruism, empathy, respect for autonomy, and holism. The findings of the present study have important implications for the delivery of patient care. Nurses should be periodically screened for insomnia-related attention deficit and be given training in sleep hygiene to improve social cognitive health which will ultimately improve their ability to provide good patient care. Future studies may target the converse, *i.e.*, amelioration in insomnia symptoms, possibly through interventions, and its impact on attention deficit and explicit social cognition. Therefore, further exploration of the study's findings may help develop strategies to manage and possibly improve healthcare delivery.

### Funding
The research was funded by the Deputyship for Research & Innovation, Ministry of Education in Saudi Arabia through the project number (lFP-2020-47). The funders had no role in study design, data collection and analysis, decision to publish, or preparation of the manuscript.

### Grant Disclosures
The following grant information was disclosed by the authors:
Research & Innovation, Ministry of Education in Saudi Arabia: lFP-2020-47.

### Competing Interests
Faizan Z. Kashoo is an Academic Editor for PeerJ.

### Author Contributions
- Md. Dilshad Manzar conceived and designed the experiments, performed the experiments, analyzed the data, prepared figures and/or tables, authored or reviewed drafts of the article, and approved the final draft.
- Faizan Kashoo conceived and designed the experiments, analyzed the data, prepared figures and/or tables, authored or reviewed drafts of the article, and approved the final draft.
- Abdulrhman Albougami conceived and designed the experiments, performed the experiments, analyzed the data, prepared figures and/or tables, authored or reviewed drafts of the article, and approved the final draft.
- Majed Alamri conceived and designed the experiments, performed the experiments, analyzed the data, authored or reviewed drafts of the article, and approved the final draft.
- Jazi Shaydied Alotaibi conceived and designed the experiments, performed the experiments, analyzed the data, prepared figures and/or tables, authored or reviewed drafts of the article, and approved the final draft.
- Bader A. Alrasheadi conceived and designed the experiments, performed the experiments, analyzed the data, prepared figures and/or tables, authored or reviewed drafts of the article, and approved the final draft.
- Ahmed Mansour Almansour conceived and designed the experiments, analyzed the data, authored or reviewed drafts of the article, and approved the final draft.
- Mehrunnisha Ahmad conceived and designed the experiments, analyzed the data, authored or reviewed drafts of the article, and approved the final draft.
- Mohamed Sherif Sirajudeen conceived and designed the experiments, performed the experiments, analyzed the data, authored or reviewed drafts of the article, and approved the final draft.
- Mohamed Yacin Sikkandar conceived and designed the experiments, analyzed the data, authored or reviewed drafts of the article, and approved the final draft.

## PeerJ

- Mark D. Griffiths conceived and designed the experiments, analyzed the data, authored or reviewed drafts of the article, and approved the final draft.

## Human Ethics

The following information was supplied relating to ethical approvals (*i.e.*, approving body and any reference numbers):

The research proposal was approved by the Human Ethics Committee, Ministry of Health, Saudi Arabia (Approval no. 9; H-05-FT-083).

## Data Availability

The raw data is available in the Supplemental File.

## Supplemental Information

Supplemental information for this article can be found online at http://dx.doi.org/10.7717/peerj.15508#supplemental-information.

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
