# Peer review of "The mediating role of attention deficit in relationship between insomnia and social cognition tasks among nurses in Saudi Arabia: A cross-sectional study"

_PeerJ, doi:10.7717/peerj.15508_

## Round 0.1 · original submission · Major Revisions

With the Reviews in hand, I recommend a major revision. I concur with the reviewers that the study investigates an important topic. However, there are a few issues that require attention/resolution. Please ensure that you satisfy the reviewers in your revisions. Examples include the need to:

Clarify the directionality of variables in the Introduction, and strengthen the theoretical explanation relating to the study variables.

Establish the validity of the single-item scale.

Include further details with regards to recruitment and the inclusion of covariates.

Specify the imputation method used.

Add further justification for the use of mediation within cross-sectional research.

Also, please ensure that you carefully go through the paper, removing repetitive/redundant sentences, unnecessary details, repetitious phrases, and short sentences with lack of clarity. One reviewer in particular identified these issues.

To conclude, I strongly encourage the authors to carefully go through the reviewers' comments, as they provide constructive suggestions for improving your manuscript. I second the request for data and script sharing for reproducibility.

Reviewer 1 ·

Basic reporting

This is an interesting study examining the mediating role of attention deficit in the association between insomnia and social cognition tasks among nurses. The paper is well-written with decent sample size. I only have a few comments to improve the manuscript further:

1. I appreciate the fact that the authors reviewed previous studies on the relationship between insomnia, social cognition and attention deficit. But it is still unclear why attention deficit is the mediator instead of the other two variables (insomnia and attention deficit). The directionality is not well-established yet in the introduction.

2, It will be important for the authors to share their data and script to ensure reproducibility of their findings. This will be useful for future meta-analysis too and potentially increase the impact and citation count of this study.

Experimental design

3. I have a major concern on the use of single-item self-reported question to measure attention deficit. I don't think it is valid at all. There is no justification and this is vulnerable to many biases. The authors should establish the validity of this scale before using it for research

4. The inclusion and exclusion criteria of the recruitment procedure should be highlighted in the method section. How were the participated recruited?

5. There is a need for a proper justification for all the covariates included

Wysocki, A. C., Lawson, K. M., & Rhemtulla, M. (2022). Statistical control requires causal justification. Advances in Methods and Practices in Psychological Science, 5(2), 25152459221095823.

Validity of the findings

7, The current study uses mediation model in a cross-sectional design. This is quite problematic and requires more discussion.

Reviewer 2 ·

Basic reporting

Thank you for this interesting manuscript.
I read it carefuly and I found it very interesting. However, The main limitation of this manuscript is the way of writing. The manuscript (especially the introduction and the methods) are too long with multiple redundant sentences(especially in the results: i.e: most of your sentences were written as: ...were significantly negativeley.....) and unecessary details that make the manuscript complexe to read in some of its parts. Also, you used multiple "orphan"sentences (i.e:1. NursesÂ’ working conditions, their sleep, and mental health conditions are often related. 2. Therefore, nurses form an eligible target population to investigate the relationship between insomnia-related attention deficit, and its correlates.3.....) and you tended to use short sentences with limited efforts to summarize.
Revise please

Experimental design

The methods needs to be summarized

Validity of the findings

Inversely, the conclusion is the less deelpped part of the manuscript

Additional comments

Title: add a term showing that the study is a cros-sectional survey
Abstract: You need to add the study period (and in the main text).
You need also to add the most important descriptive features of te population
Keywords: add insomnia, attention problems
Introduction: Line 58: Extended and varying shifts are (what type of shifts?)
ADHD: Add the complete definition in its first apparition (then you can use the abbreviation)
The discussion needs to be revised. It seemed to b a second litterature review than a discussion. Try to explain some information please

Reviewer 3 ·

Basic reporting

A cross-sectional study surveyed 664 nurses to measure their prevalence of insomnia symptoms, attentional deficit, and attitude towards patient. While insomnia symptoms significantly correlated with displays of poor attitude towards patient, the correlation was accounted for by the significant indirect effects of attentional deficit on attitude towards patient. More specifically, on each dimension of the attitude scale, which measures also respect for patient’s autonomy, holistic attitude towards patients, empathy with patients, and altruistic behavior towards patients, a similar pattern of result was observed; while insomnia symptoms significantly correlated with poorer outcomes on these dimensions, the correlation was explained by the indirect effect of attentional deficit on the attitude dimensions. In sum, attentional deficit was shown to mediate the relationship between insomnia and poorer attitudes towards patients via the mediation by measurement model.
The study addresses a crucial population of nurses and a practical issue of patient care in the healthcare industry. The methods of the study were in general sound and well justified. Ideally, the authors can take into consideration some of the comments detailed below.

1. This is a minor comment, but I suspect the wrong citation has been provided in line 187, that provides a reference for the past use of the single-item attention deficit scale. The citation provided brings to a study done on another population of nurses and not university students.

Experimental design

2. While the authors presented evidence from previous research to demonstrate the possible links between their three key variables: Insomnia, attention deficit, and social cognition, there does not seem to be strong theoretical explanation behind why these links may be present and important. For instance, between insomnia and attention deficit, a question of how sleep may impact attention deficit is important to address beyond associating insomnia with the ADHD population. In fact, using ADHD symptoms to inform the possible mechanisms behind insomnia leading to attention deficit seems counterintuitive as ADHD may be pre-disposed and diagnosed even in developmental/childhood stages in that rather than insomnia predicting ADHD, it is more likely to infer ADHD predicts insomnia. However, the direction of influence the authors seem to focus on in this paper is the opposite; insomnia should be a factor of attention deficit. What I might suggest is for the authors to consider leveraging on the attentional control literature to bolster the theoretical link between insomnia and attention deficit. Perhaps see Lim & Dinges (2008) for a relevant work on sleep deprivation and vigilant attention.

3. Building on the previous comment, in drawing links between insomnia and social cognition, as well as attention and social cognition, the authors seemed to place heavy focus on the construct of empathy to represent social cognition (lines 95-117). However, their measure of social cognition is an attitudinal scale that fully encompasses other dimensions besides empathy, such as respect for autonomy, altruism, and holistic care. It feels as if only the empathy dimension has received attention in the introduction and not much was done to explain why the authors suspect these other dimensions to be impacted by insomnia and attention deficit.

Validity of the findings

4. It is unclear what iterative imputation method has been used by the authors to account for missing data. Does multiple imputation refer to the multiple imputation by chained equations (MICE) method? It might be beneficial for the authors to clarify the method used as different imputation methods may have different implications on missing data and the analysis which comes after. I generally refer to Liu & Brown (2013, CILS) for a better idea of the implications of each imputation method.

Additional comments

The paper was well-structured and presented important findings implicating the healthcare industry and patient-care optics. Although there is some need to better clarify the theoretical justification for the proposed links and mechanisms of the constructs present in this paper, it is apparent that the authors have put in great effort in the discussion to round off the edges. The introduction however, may benefit from more thorough explanations of the rationale behind the probable links

---

## Round 0.2 · Minor Revisions

With the Reviews in hand, I recommend a minor revision.

I concur with the reviewers that the manuscript is an important piece of work, but it also lacks some important details, and there are grammatical issues in places.

The reviewers also pointed out that the Introduction and Discussion sections require more improvement.

To conclude, I strongly advise the authors to carefully go through the reviewers' comments, as they both provided constructive suggestions for improving your manuscript.

Reviewer 1 ·

Basic reporting

The authors have sufficiently addressed my comments

Experimental design

The authors have sufficiently addressed my comments

Validity of the findings

The authors have sufficiently addressed my comments

Reviewer 2 ·

Basic reporting

I would thank the authors for their efforts to improve the qualtiy of the manuscript. However, more efforts are required:
1. Even the authors have made efforts to review and revise the introduction, little is done and the summarising efforts cannot be seen (except some sentences reformulations).
2. The same for the discussion where any effort was done to revise and/or to summarize.

Experimental design

The methods are stiil too long and no efforts were done to sumarize

Validity of the findings

.

Additional comments

Please add the country name to the title
Be careful to some errors in the references : exp: Line 55: (organisation, 2016),
Be carul also the some spelling errors related to number exp: in the abstract: 0.18 but .56 (unify in alm the manuscript please
Also is it correct to find a negative CI? please confoirm and revise?

Reviewer 3 ·

Basic reporting

There are still minor grammatical errors present in the manuscript. The paper will benefit from a more thorough round of proof-reading. Some examples of grammatical error:
1. Lack of proper punctuations and spaces for "Sleep loss leads to erroneous emotion differentiation (Simon et al., 2020)Individual-level emotional processing coupled with attentional bias and/or decreased vigilant attention..." (p. 5, lines 7-9).
2. For line "...insomnia lead to decreased vigilant attention, and attentional bias, that may express in problems in attitude and social cognition" (p.6, lines 2-3). The phrase "express in problems in" seems slightly awkward. I wonder if a better phrasing would be "express as problems in". This is for the authors to consider.

Experimental design

No comment. The authors did well to address the concerns of all reviewers.

Validity of the findings

No comment. The authors did well in justifying the use of the single-item measure of attention deficit.

Additional comments

In general, the authors addressed my previous comments well. The theoretical foundation behind the hypothesis of the paper has been strengthened.

---

## Round 0.3 · accepted · Accept

I am glad to inform you that the remaining concerns expressed by the Reviewers have now been addressed and, therefore, the manuscript is suitable for publication.